# B Lymphocyte Development in the Bursa of Fabricius of Young Broilers is Influenced by the Gut Microbiota

Jiaheng Cheng,[a] Huangtao Lei,[a] Chunlin Xie,[a] Jianwei Chen,[b] Xinyu Yi,[b] Fang Zhao,[b] Yushan Yuan,[a] Peng Chen,[a] Jingyi He,[a] Chenglong Luo,[a] Dingming Shu,[a] Hao Qu,[a] ⓘ Jian Ji[a]

[a]State Key Laboratory of Livestock and Poultry Breeding, Guangdong Key Laboratory of Animal Breeding and Nutrition, Institute of Animal Science, Guangdong Academy of Agricultural Sciences, Guangzhou, China

[b]BGI-Shenzhen, Shenzhen, China

**ABSTRACT** Chickens have been used as a valuable and traditional model for studies on basic immunology. B lymphocytes were first identified in the bursa of Fabricius (BF) of broilers. The microbiota is important for immune system development and function. However, the effect of the microbiota on mediating B cell development and its regulatory mechanism is poorly elucidated. Here, we show that the gut microbiota is associated with the development of bursal B cells in young chickens. Changing patterns of both the alpha diversity and the expression of the B cell marker Bu-1$\alpha$ in the gut microbiota were related to the ages of chickens at different growth phases. Further correlation analysis revealed the marked correlation between the relative abundances of *Intestinimonas*, *Bilophila*, *Parasutterella*, *Bacteroides*, *Helicobacter*, *Campylobacter*, and *Mucispirillum* and the expression of Bu-1$\alpha$. In antibiotic-treated chickens, BF and B cell development had aberrations as the relative abundance of the microbiota in early life decreased. These findings were consistent with Spearman's correlation results. Single-cell transcriptome analysis indicated that the heterogeneity in the cellular composition and developmental trajectory of bursal B cells from antibiotic-treated chickens was large. We found a novel subpopulation of unnamed B cells and identified Taf1 as a new pivotal regulator of B cell lineage differentiation. Therefore, we provide novel insights into the regulatory role of the gut microbiota in B cell development in early life and the maturation of host humoral immunity.

**IMPORTANCE** In this study, we used young broilers to investigate the relationship between their gut microbiota and bursal B cell development. We characterized the important variables, microbes, B cells, and immunoglobulins during the posthatch development of birds. We also identified several candidate taxa in the cecal contents associated with B cells. Our study provides a rich resource and cell-cell cross talk model supporting B cell differentiation from the bursa *in vitro* at single-cell resolution. Furthermore, we determined a new pivotal regulator (Taf1) of B cell differentiation. We believe that our study makes a significant contribution to the literature because our findings may elucidate the role of the gut microbiota in B cell differentiation. This study also serves as a basis for developing new strategies that modulate B cell differentiation to prevent diseases.

**KEYWORDS** B lymphocyte, bursa, gut microbiota, broiler, early life

Address correspondence to Hao Qu, quhao@gdaas.cn, or Jian Ji, jijian1017@163.com.

The authors declare no conflict of interest.

B cells were first described in the bursa of chickens by Cooper et al. (1). They develop in a microenvironment in the bursa of Fabricius (BF), which is unique to birds (1). Interestingly, the bursa undergoes age-dependent alterations; thus, its size changes during development, and it rapidly grows during late embryogenesis and subsequently regresses after sexual maturation (2, 3). The bursa of birds is connected to the cecum, which is an area of the large intestine that contains most of the gut microbiota,

via a duct. Therefore, the duct may indicate a relationship between the gut microbiota and bursal B cells in birds.

The gut microbiota plays a fundamental role in the induction, education, and function of the host immune system (4). Germfree (GF) animals exhibit major defects in thymus development (5). Dysbiosis of the neonatal gut microbiome induces the dysfunction of CD4$^+$ T cells, which is associated with childhood atopy (6). The number of regulatory T (Treg) cells, which can be normalized through standard conventionalization and mono-colonization with certain *Clostridium* species or various intestinal microbes, is decreased in the GF mouse colon (7, 8). Our previous study showed that commensal microbe-derived butyrate facilitates the polarization of M2 macrophages, which play a critical role in dextran sulfate sodium-induced colitis (9). GF mice exhibit a general defect in the production of IgA and IgG antibodies in mucosal and nonmucosal organs, which can be normalized after conventionalization by the gut microbiota (10, 11). Similarly, antibiotic (ABX)-treated chickens had a significantly greater decrease in the number of macrophages than the controls in early life (12). The gut microbiota of chickens can modulate immune responses and protect against enteric pathogens (13).

Studies have yet to determine whether the gut microbiota is related to the bursa in chickens, and the underlying mechanisms remain unknown. Here, we provide evidence that the gut microbiota plays a critical role in B cell development in the bursa of chickens in early life. We also reveal critical and previously unappreciated information for understanding the pathophysiology of immune system diseases.

## RESULTS

**Developmental size and structural changes in the BF.** To assess the development of the BF in chickens, we observed the changes in the size and weight of the BF by a time course analysis. The results showed that the shape, size, and weight of the BF increased from days 1 to 112 and sharply declined thereafter (Fig. 1A and B). The ratio of the BF to the body weight peaked at approximately 21 days of age and then decreased gradually with age (Fig. 1C). For the other immune organs, the changes in splenic shape, size, and weight with age were slower and smaller than those in the BF of birds after 112 days (see Fig. S1A to C in the supplemental material).

The BF on day 9 attained a more mature histological structure with an increased size of lymphoid follicles and a clear medulla and cortex compared with those on day 1 (Fig. 1D and E and Table 1). From day 9 to day 70, the lymphoid follicle size markedly increased (Fig. 1F to H and Table 1) and exhibited a stable trend until day 112 (Fig. 1H and Table 1). The lymphocyte density and number and the size of the lymphoid follicles in the BF decreased on days 140 and 154 compared with those in the birds in the other age groups. Additionally, fibrous tissues accumulated, and mucoid cysts enlarged as the cortical and medullary thickness of the BF decreased (Fig. 1I and J, Table 1, and Table S1), indicating that the bursa was undergoing involution.

Collectively, these results indicated that the size and structure of the BF developed with the growth of the broiler, while when the chickens were mature, these parameters regressed.

**Gut microbiota associated with the development of the expression of Bu-1$\alpha$, IgA, IgM, and IgY in the BF.** B cells were found principally in the cortex and medulla of the bursal follicles. The expression levels of Bu-1$\alpha$ (a marker for B cells) and three immunoglobulins (IgY, IgM, and IgA) in the bursa at various developmental stages were examined by immunohistochemistry. At earlier stages, the expression levels of Bu-1$\alpha$, IgY, IgM, and IgA in the bursal follicles gradually increased until day 49 and became stable on day 70 and day 112. Their expression gradually disappeared, and almost all of them were undetectable in the bursal follicles on day 140 and day 154 (Fig. 2A to D and Fig. S2 to S5). The expression patterns in the cortex and medulla of follicles were similar; the expression levels of Bu-1$\alpha$ and the three immunoglobulins increased sharply from day 1 to day 49 and peaked from day 49 to day 112. On day 140 and day 154, the cortex and medulla could not be differentiated because of follicle cavitation and shrinkage (Fig. S2 to S6).

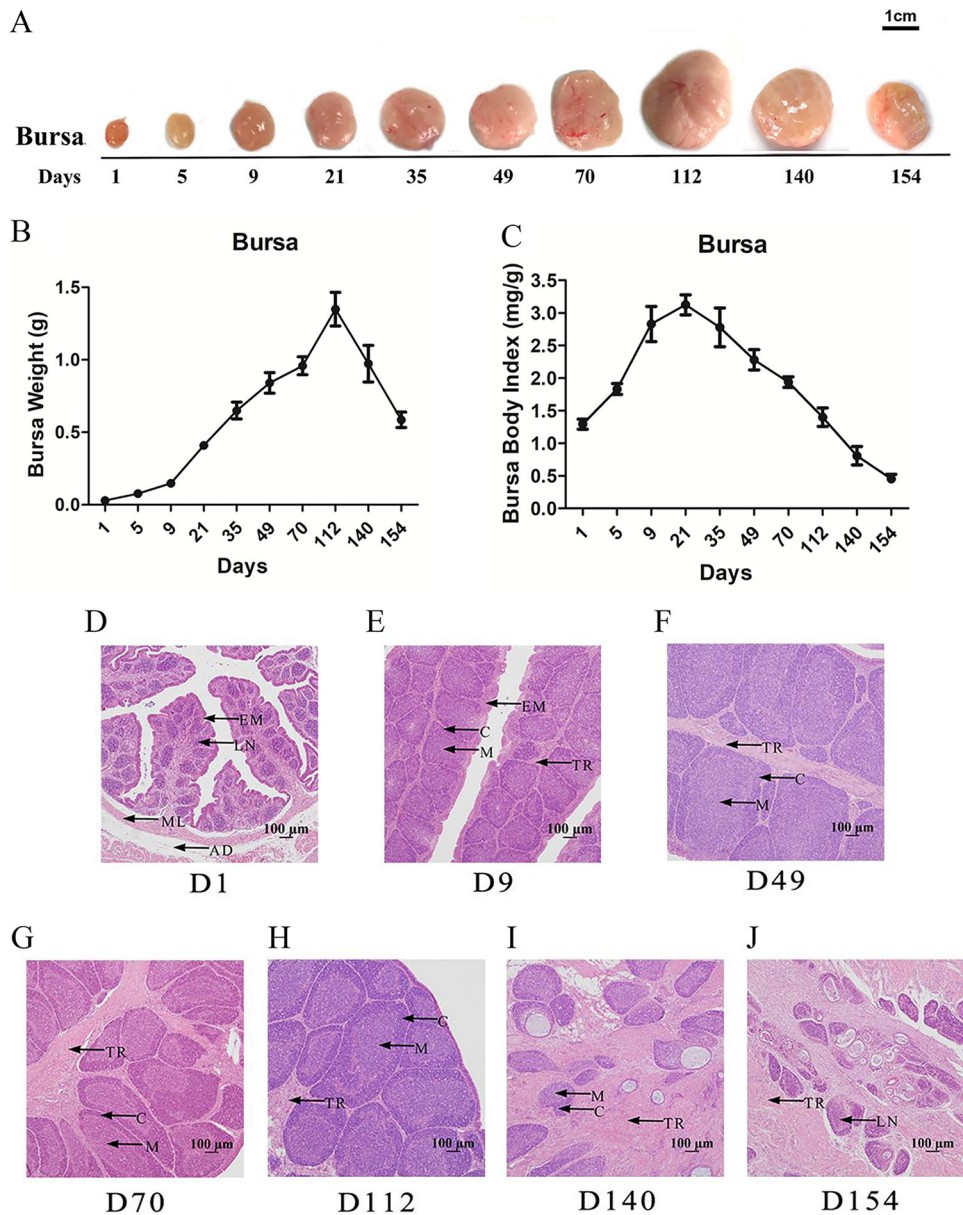

**FIG 1** Size, weight, and histological structure of the bursa of Fabricius (BF) at different ages. (A) Size of the BF from days 1 to 154. (B) Weight of the BF from days 1 to 154. (C) Ratio of the BF to the body weight from days 1 to 154. Data are means ± standard errors of the means (SEM) ($n = 6$). (D to J) BF on day 1 (D1) (D), day 9 (E), day 49 (F), day 70 (G), day 112 (H), day 140 (I), and day 154 (J). LN, bursa lymph nodule; EM, epithelium mucosa; ML, muscular layer; AD, tunica adventitia; M, medulla; C, codex of the lymphoid follicle; TR, trabecula ($n = 6$).

To further understand the relationship between the gut microbiota and bursal B cells, we studied the changes in the gut microbiota of birds of different ages. Three alpha diversity scores, the Chao1 index, the number of observed species, and the Shannon index, substantially increased from day 1 and continued to increase until day 49. These scores were stable during adulthood from day 70 to day 154 (Fig. 2E to G). We further assessed the intersample variability in the community structure (beta diversity) through unsupervised principal-coordinate analysis (PCoA) of Bray-Curtis dissimilarity. PCoA clearly showed statistical differences in the community structures on day 1, day 9, and day 49, but no clear clusters were observed between day 49 and day 70. The microbiota on day 49 and day 70 clustered separately from those of the three other stages (day 112, day 140, and day 154) (Fig. 2H). Spearman's correlation

**TABLE 1** Histometric measurements of the bursa of Fabricius at different ages[a]

| Age (days) | Mean cortical area of the BF ($\mu m^2$) ± SE | Mean nodular area of the BF ($\mu m^2$) ± SE | Mean medullary area of the BF ($\mu m^2$) ± SE |
|---|---|---|---|
| 1 | 6,031 ± 682 D | 13,436 ± 1,617 G | 8,120.4 ± 1,130 D |
| 9 | 14,689 ± 1,617 C | 51,530.18 ± 5,679 E | 17,776.5 ± 969 C |
| 49 | 77,702 ± 4,823 A | 153,084.9 ± 11,744 C | 59,027.6 ± 6,102 A |
| 70 | 61,355 ± 9,769 A | 159,796.8 ± 12,891 C | 54,109.6 ± 8,894 A |
| 112 | 61,457 ± 5,472 A | 160,555.5 ± 7,088 C | 49,830.8 ± 4,598 A |
| 140 | — | 68,422.8 ± 5,610 B | — |
| 154 | — | 25,629.1 ± 1,839 A | — |

[a]Data are means ± standard errors ($n = 6$). Different letters in the same column indicate significant differences ($P < 0.05$). —, unmeasurable.

analysis was performed to further explore the relationship between the gut microbiota and bursal B cells. At the phylum level, with correlations exceeding 0.6, *Bacteroidetes* and *Deferribacteres* were positively related to the expression of Bu-1$\alpha$, IgA, IgY, and IgM (Fig. 2I). At the genus level, with correlations exceeding 0.6, *Intestinimonas*, *Bilophila*, *Parasutterella*, *Bacteroides*, *Helicobacter*, *Campylobacter*, and *Mucispirillum* were positively correlated with Bu-1$\alpha$, IgA, IgY, and IgM (Fig. 2J).

In summary, the expression levels of Bu-1$\alpha$ and the three immunoglobulins in bursal follicles gradually increased in early life until day 112 and decreased thereafter. Furthermore, changes in B cells and the gut microbiota were related to chicken age. The patterns of changes in the alpha diversity of the gut microbiota and the expression of the B cell marker Bu-1$\alpha$ in chickens of all ages were similar on different days.

**Antibiotic-induced microbiota depletion affects B cells and the expression of IgA, IgM, and IgY in the bursa.** Broad-spectrum antibiotics (ABX) were used to eliminate the gut microbiota in birds and further evaluate the cross talk between the gut microbiota and B cells in the bursa. These results showed that antibiotic treatment did not alter ($P > 0.05$) the size and weight of the BF or the ratio of the BF to the body weight over time (Fig. S7A to C). Additionally, no effect ($P > 0.05$) was observed in the nodule area of the bursa (Fig. S8A and B).

Bu-1$\alpha$ expression was assessed to investigate the changes in the proportions and numbers of bursal B cells during the developmental stages by flow cytometry. The numbers and percentages of Bu-1-positive (Bu-1$^+$) cells in the bursa were evaluated during antibiotic treatment. The results showed that B cells were the predominant cell type in the bursa. The percentage of Bu-1$^+$ cells in the bursa of the treated birds was not significantly different from that of the controls on day 1, day 9, and day 140 ($P > 0.05$) (Fig. 3A and B), but it tended to be lower on day 49 than on the other days (41.98% ± 6.27% for the controls and 36.86% ± 9.02% for the antibiotic treatment). The number of Bu-1$^+$ cells in the treated birds on day 49 was significantly lower ($P < 0.05$) than that in the controls (Fig. 3C). The levels of immunoglobulins in the bursa showed no significant differences ($P > 0.05$) between the birds treated with antibiotics and the controls on day 1 and day 9. On day 49 and day 140, the levels of IgY, IgA, and IgM decreased ($P < 0.05$) in the bursa compared with those in the controls (Fig. 3D to F and Fig. S9).

Antibiotic treatment was conducted to further investigate the changes in the gut microbiota and examine the specific gut microbiota profile associated with B cells. As expected, antibiotic treatment dramatically reduced the stool alpha diversity on day 9, day 49, and day 140 (Fig. 3G). PCoA showed statistical differences between the antibiotic and control groups on day 9, day 49, and day 140, especially on day 9 (Fig. 3H). At the phylum level, with a positive immunoglobulin correlation of >0.6, the relative abundances of *Bacteroidetes* and *Deferribacteres* decreased on day 49 and day 140 during antibiotic administration (Fig. 3I). At the genus level (Fig. 3J), seven genera had positive immunoglobulin correlations of >0.6: *Mucispirillum*, *Campylobacter*, *Helicobacter*, *Bacteroides*, *Parasutterella*, *Bilophila*, and *Intestinimonas* ($P < 0.05$). Interestingly, the abundances of several taxa and genera decreased, which agreed with our above-described observations by Spearman correlation analysis (Fig. 2I and J). The relative abundances of these seven genera decreased on different days. *Intestinimonas* started decreasing on day 9; *Bilophila*,

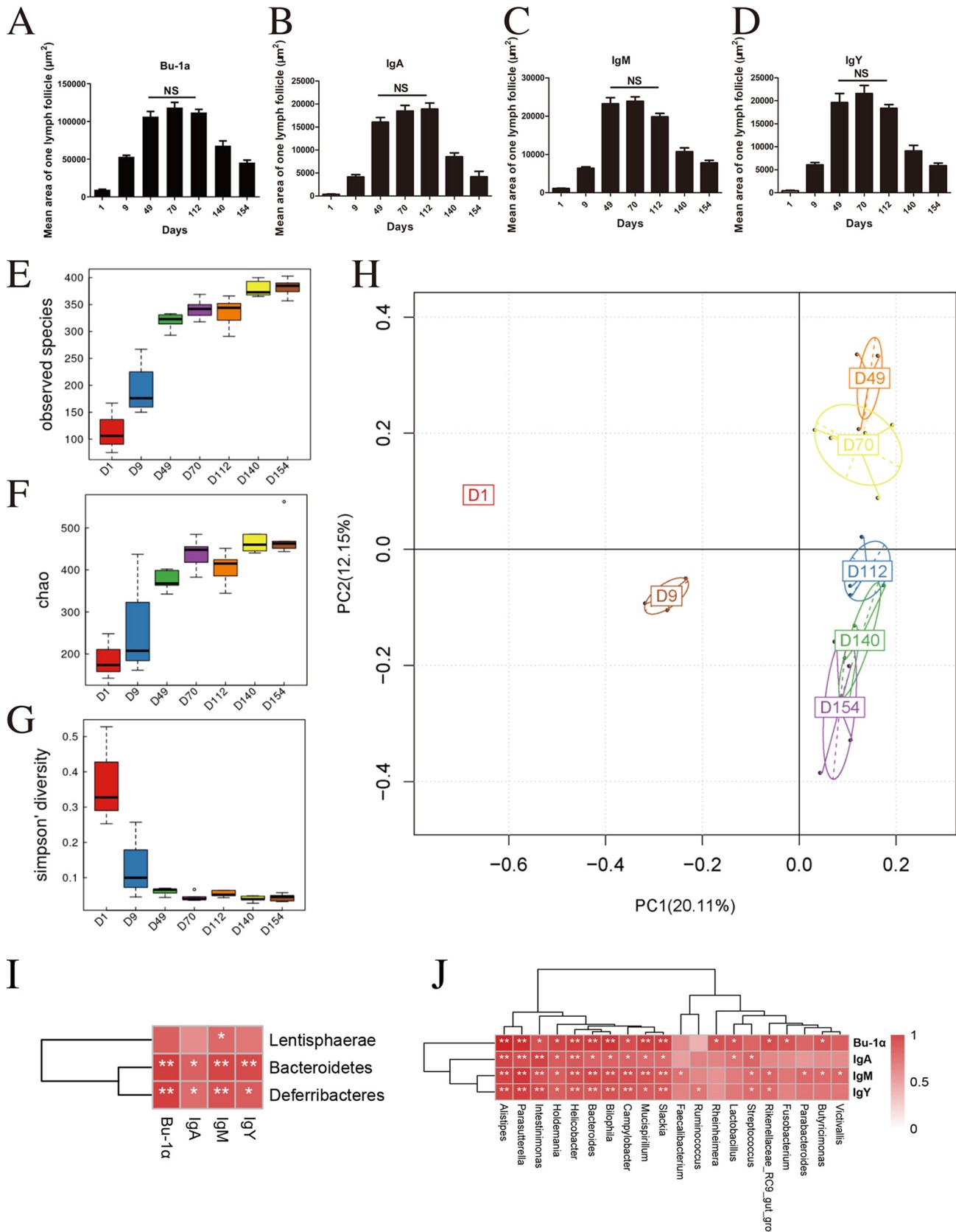

**FIG 2** Expression of Bu-1α, IgA, IgM, and IgY is associated with the gut microbiota during the development of the bursal B cell. (A) Bu-1α-positive lymphoid follicle. (B) IgA-positive lymphoid follicle. (C) IgY-positive lymphoid follicle. (D) IgM-positive lymphoid follicle on different days. Ten random

*Mucispirillum*, and *Bacteroides* started decreasing on day 49; and *Helicobacter* and *Campylobacter* started decreasing on day 140 ($P < 0.05$).

In summary, the number of bursal B cells substantially decreased at earlier stages (before day 49) of antibiotic treatment, suggesting that it was a critical period for the gut microbiota-mediated elevated production of B cells in the bursa. These results further confirmed that the gut microbiota influenced or likely regulated B cell production in the bursa of birds in early life.

**Single-cell transcriptomic analysis identifies bursa cell populations.** Bursa cell types are typically identified by single-cell RNA sequencing (scRNA-seq). In this study, we obtained single-cell transcriptomes for 27,337 cells in the samples, 76,735 in the controls, and 92,968 in the antibiotic-treated chickens. We combined 197,040 cells for cluster analysis and identified 22 clusters (Fig. 4A). We identified B cell populations by using the marker gene *CHB6*. Among them, 16 clusters were B cells, and 6 clusters were non-B cells (Fig. 4B and C). To more accurately dissect B cell subpopulations, we further classified them into large and small B cells (Fig. 4D). Consistent with the developmental function of large B cells in the bursa, the mRNA expression levels of the B cell differentiation-related genes *CXCR4*, *BLNK*, *PAX5*, *IKZF3*, *IKZF1*, *EBF1*, *FOXO1*, and *TNFRSF13C* were higher in large B cells than in small B cells (Fig. 4H). The percentage of bursal B cells showed no significant differences between the birds treated with antibiotics and the controls (Fig. 4E to G). These analyses confirmed the strong agreement between the patterns observed by scRNA-seq and protein levels determined by flow cytometry (Fig. 3B).

In conclusion, the scRNA-seq data set contained 16 clusters of bursal B cell types, which could be further divided into at least two B cell subpopulations with distinct developmental potentials *in vivo*.

**Reconstructing the developmental trajectory of bursal B cells.** To further understand the relationship between different types of bursal B cells and their developmental trajectory, we performed Monocle pseudotime trajectory analysis in which individual cells were aligned along a developmental trajectory. Interestingly, these 16 B clusters (three states) had the following developmental order: large B cells → small B cells and unnamed B cells (Fig. 5A). A branched point from the linear path of the pseudotime trajectory was observed, demonstrating that one branched group of cells represented the differentiation from large B cells to small B cells; the other branched group of cells represented unnamed B cells (Fig. 5A). We found that antibiotic treatment significantly altered the developmental trajectory and differential gene expression patterns of large B cells, small B cells, and unnamed B cells (Fig. 5B and Fig. S10). To investigate the mechanism of B cell differentiation retardation under antibiotic treatment, we performed Gene Ontology (GO) analysis to show that distinct categories of differentially expressed genes (DEGs) were enriched in the control and antibiotic treatment groups. DNA packaging, chromatin assembly or disassembly, nucleosome organization, chromatin assembly, and nucleosome assembly were highly enriched GO terms of B cell clusters in the control group versus the ABX group (Fig. 5C). Therefore, ABX might significantly alter the epigenetic states of nuclear and chromatin assembly during B cell differentiation (Fig. 5C).

**Expression patterns of transcription factors in bursal B cells.** We used SCENIC (single-cell regulatory network inference and clustering) to identify transcription factors (TFs) in different bursa B cell subpopulations and predict the essential regulators of B cell types. We further applied SCENIC to investigate the transcription factors that might regulate bursa B cell differentiation. Our network analysis identified the top 3 specific regulons (Taf1, Ets1, and Elk3) in large and small B cells (Fig. 6A). To investigate

**FIG 2** Legend (Continued)

fields per specimen were analyzed with Image Pro-Plus V.6, and the mean fluorescence intensity per power field was recorded. Data are means ± SEM (NS, not significant) ($n = 6$). (E to G) Three alpha diversity scores based on the Chao1 index, the number of observed species, and the Shannon index. (H) Principal-coordinate analysis of cecal microbial beta diversity based on Bray-Curtis dissimilarity for each age group ($n = 6$). (I and J) Spearman correlations among the gut microbiota, Bu-1$\alpha$, and three immunoglobulins at the phylum (I) and genus (J) levels. Spearman's correlation coefficient was calculated between the microbial composition and Bu-1$\alpha$, IgA, IgY, and IgM. A $P$ value of $<0.05$ and a correlation of $>0.6$ were considered statistically significant ($n = 6$). *, $P < 0.05$; **, $P < 0.01$.

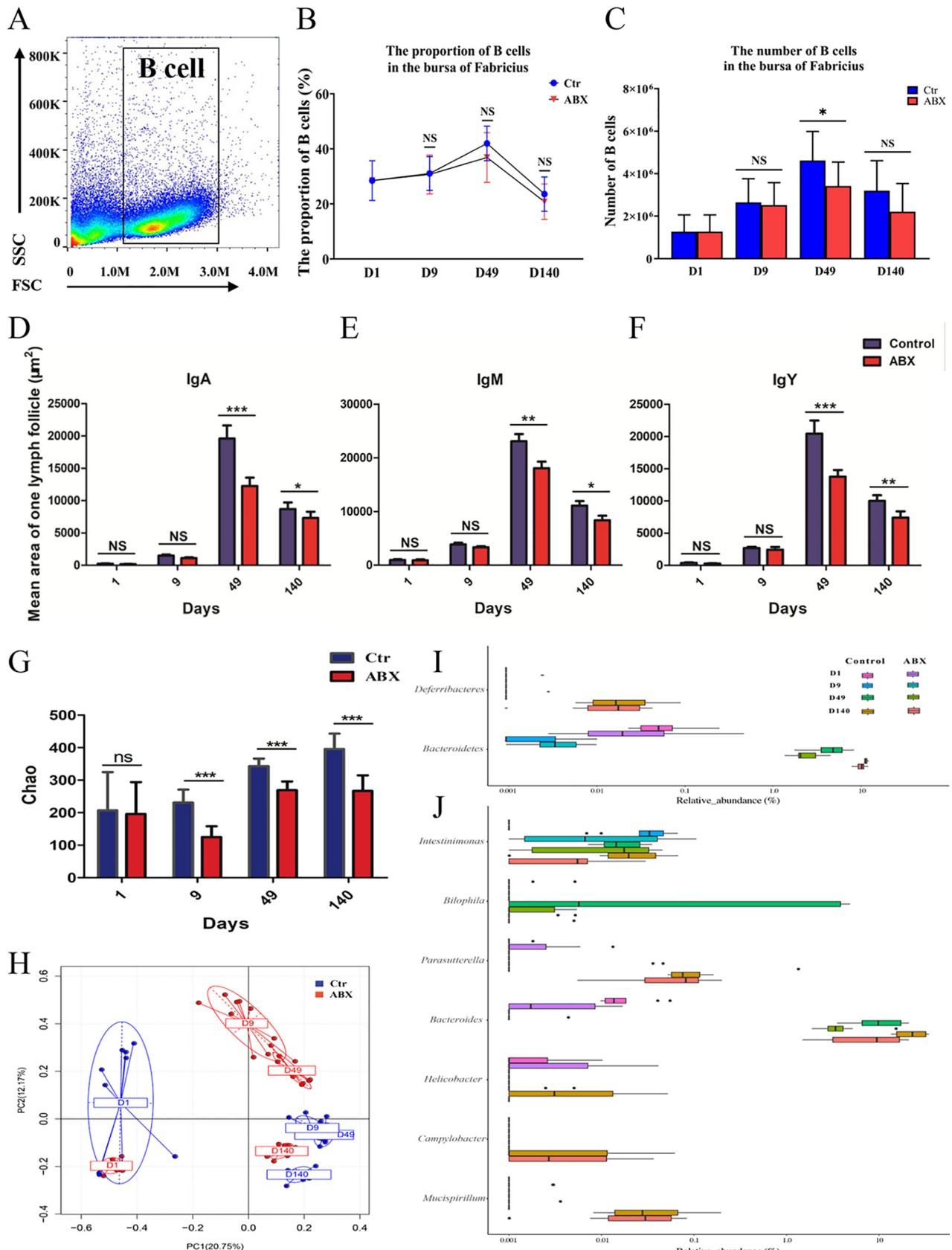

**FIG 3** Changes in Bu-1α, IgA, IgM, and IgY expression in the bursa of Fabricius and the gut microbiota with antibiotic treatment. (A) Single cells produced from the bursa were stained with an anti-Bu-1 antibody. FSC, forward scatter; SSC, side scatter. (B) The percentage of bursal B cells was

the function of these three regulons, we searched for their target genes by using the KEGG pathway database. The target genes of these three regulons participated in cellular, metabolic, biological, developmental, and immune system processes (Fig. 6B). We focused on the identification of DEGs related to the immune system between the controls and ABX birds because B cells play a central role in adaptive immunity. We identified 199 DEGs, including 23 upregulated and 13 downregulated ones (Fig. 6C). Further analysis of the gene regulatory network (GRN) showed that Elk3 regulated a large number of target genes much more than Taf1 and Ets1 did. With antibiotic treatment, the number of Elk3 target genes increased from 65 to 124, and the numbers of Ets1 and Taf1 target genes decreased from 4 to 2 and from 16 to 6, respectively (Fig. 6D).

These findings suggested that Taf1, Ets1, and Elk3 changed the expression of target genes, resulting in various altered cellular properties that contributed to differentiation.

## DISCUSSION

In this study, we identified candidate members of the gut microbiota associated with B cell development in the BF. Although the members of the gut microbiota did not directly alter the size and weight of the BF, they affected the proportion and number of bursal B cells. Additionally, continuous antibiotic treatment decreased the expression levels of IgA, IgM, and IgY in the bursa. Furthermore, we performed single-cell RNA sequencing to establish a high-resolution map of bursal B cells and identified a potential novel transcription factor, Taf1, involved in B cell differentiation.

Bursa-dependent B cells are solely responsible for antibody production in birds. The chicken bursa is a useful animal model to study B cell differentiation and function because it is unique for B cell development and easily accessed anatomically on specific days (14). The BF, a gut-associated lymphoid tissue, is connected to the cloaca by a duct and is affected by the intestinal microbiota (15). In the present study, we found similar patterns of changes in the gut microbiota and B cells in chickens on the indicated days. Interestingly, the expression of three immunoglobulins and alpha diversity scores peaked on day 49. Consistent with our above-described observations by Spearman correlation analysis, the abundances of two relatively positive phyla and seven genera decreased on different days after antibiotic treatment. These results suggested that the members of the gut microbiota were involved in B cell development in the bursa.

In this study, neither the size or weight of the BF nor the nodule area of the bursa was changed after antibiotic treatment. However, the administration of broad-spectrum antibiotics reduced the levels of immunoglobulins in the bursa on day 49 and day 140. A similar result was reported in a previous study, which found that the IgY level decreases in response to amoxicillin-induced changes in the gut microbiota and serum (16). Vancomycin treatment induces a reduction in the number of Treg cells and impairs the induction of Th17 cells in the lamina propria of the colon (7). Previous studies also showed that gut microbiota alterations caused a decrease in IgA, a noninflammatory immunoglobulin involved in pathogen and allergen exclusion (17–19). Intestinal IgA has decreased responses to immature structures of Peyer's patches and isolated lymphoid follicles in GF mice (20, 21).

Antibiotic treatment was conducted to further examine the specific gut microbiota profile associated with B cells. At the phylum level, *Bacteroidetes* and *Deferribacteres* were positively correlated with immunoglobulin levels. At the genus level, for a positive immunoglobulin correlation of >0.6, seven genera were observed: *Mucispirillum*, *Campylobacter*, *Helicobacter*, *Bacteroides*, *Parasutterella*, *Bilophila*, and *Intestinimonas* ($P < 0.05$). Exposure to microbial antigens in early life could determine the clonality of

**FIG 3** Legend (Continued)

examined by flow cytometry on days 1, 9, 49, and 140. (C) The number of bursal B cells was examined by flow cytometry on days 1, 9, 49, and 140. (D) The area occupied by IgA[+] B lymphocytes in one lymphoid follicle. (E) The area occupied by IgM[+] B lymphocytes in one lymphoid follicle. (F) The area occupied by IgY[+] B lymphocytes in one lymphoid follicle. *, $P < 0.05$; **, $P < 0.01$; *** $P < 0.001$ ($n = 10$). (G) Alpha diversity scores based on the Chao1 index. (H) Principal-coordinate analysis of cecal microbial beta diversity based on Bray-Curtis dissimilarity for each age group. (I and J) Changes in the gut microbiota at the phylum and genus levels on days 1, 9, 49, and 140. Each data point is the mean with the SEM. ***, $P < 0.001$ ($n = 10$).

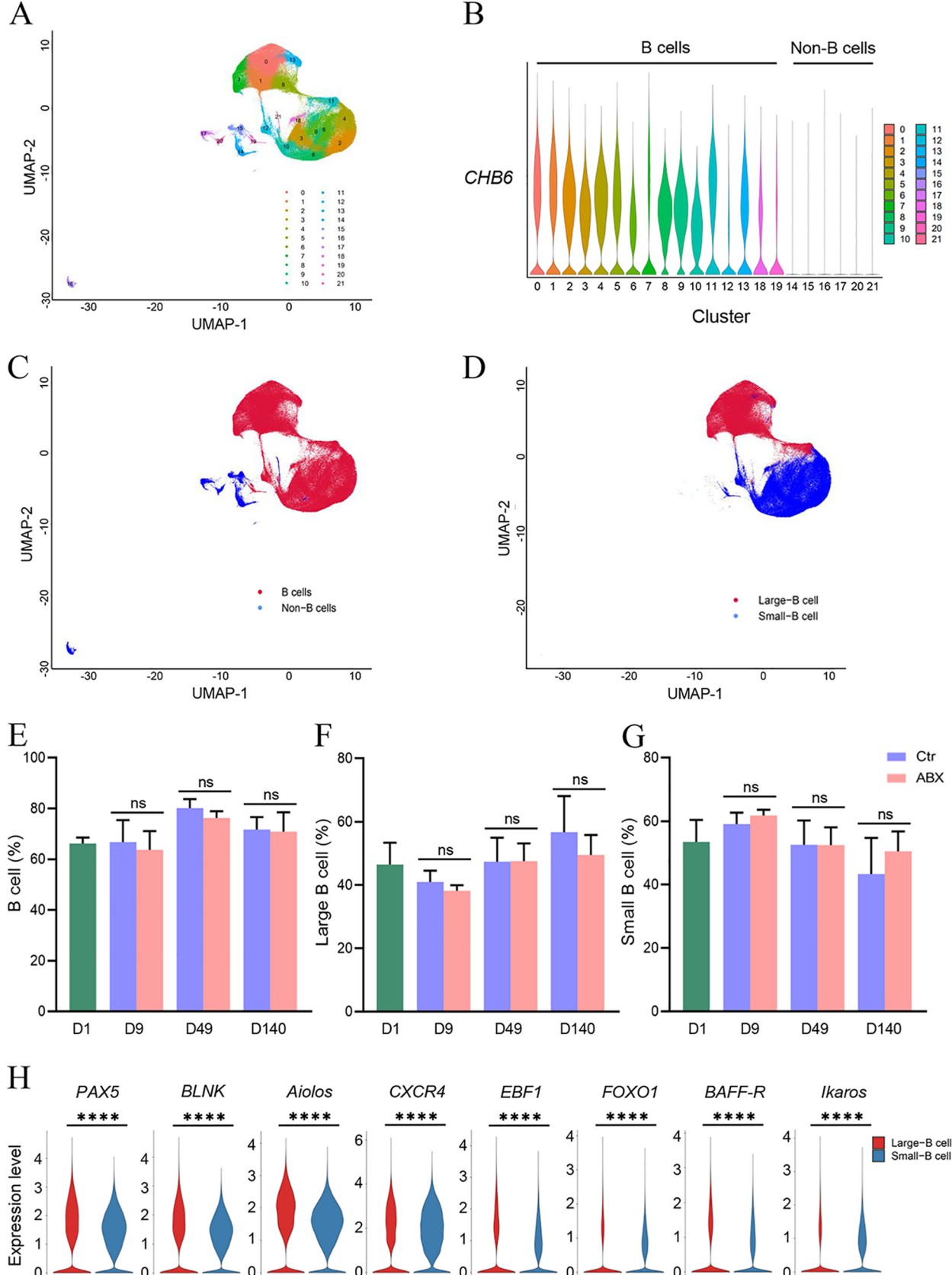

**FIG 4** Single-cell analysis of intrabursa cell populations. (A) UMAP plot showing the identified cell populations (all cells = 197,040) within 21 organ tissues from all groups merged. Each dot represents one cell, with color coding according to the origin of the organ. Labeled cell types

the mature B-1 B cell repertoire and affect antibody responses. N-Acetyl-D-glucosamine-reactive B-1 clonotypes and serum antibodies are reduced in GF mice compared with those in conventionally raised mice (22). The generation of Treg cells depends on the gut microbiota (23). For instance, *Bacteroides fragilis* and *Clostridium* species belonging to phylogenetic groups IV and XIV promote the differentiation of T cells into Treg cells in mice (7, 24). Conversely, segmented filamentous bacteria promote the differentiation of proinflammatory Th17 cells (25). Our previous studies revealed that the microbial metabolite butyrate facilitates M2 macrophage polarization and function (9). Moreover, we found that antibiotic treatment significantly reduced the total number of bursal B cells. This result suggested that the bursal B cell was likely a microbiota antigen-independent inherent feature during embryonic development in chickens.

Further identification of bursa B cell subtypes and single-cell transcriptomic mapping are necessary to understand the biology of bursa B cell development and the key role of the gut microbiota in the development of these B cells. Using single-cell RNA sequencing techniques, we classified bursal B cells into 16 clusters. Previous studies revealed two types of basal subpopulations (large and small B cells); conversely, the present study found three basal-like cell subpopulations with significantly different gene signatures. Here, we describe the transcriptome profiles of bursal B cells and highlight a novel subpopulation of unnamed B cells; further studies should be performed to identify this novel B cell subpopulation.

Our findings clearly revealed that ABX might significantly alter DNA packaging, chromatin assembly or disassembly, nucleosome organization, chromatin assembly, and nucleosome assembly, affecting B cell differentiation (Fig. 5C). Chromatin accessibility is required for B cells to differentiate into plasma cells (26). The extent of DNA accessibility varies, and nucleosome chain folding between different functional and epigenetic states of nuclear chromatin changes dramatically upon cell differentiation (27).

Our network analysis identified the top 3 specific regulons (Taf1, Ets1, and Elk3) associated with B cell differentiation in birds. Previous studies showed that Ets1 and Elk3, which affect B cell development and differentiation, are highly expressed in B cells. Ets1, a prototypical member of the *Ets* gene family, is necessary to maintain cells in a quiescent state; the loss of Ets1 leads to premature B cell differentiation into antibody-secreting cells (28, 29). Elk3, an Ets domain transcription factor, is highly expressed primarily at the early stages of B cell development; its expression decreases significantly upon B cell maturation and is associated with the activity of the enhancer of the immunoglobulin heavy chain (30). Taf1 is involved in myogenic and human embryonic stem cell differentiation (31, 32). The transcription factor reconfigures the three-dimensional chromatin architecture to control gene expression and then drives somatic cell transdifferentiation (33). Taf1 is a regulatory protein involved in the coordinate expression of polymerase I (Pol I)- and Pol II-transcribed genes required for protein biosynthesis and cell cycle progression (34). To our knowledge, this study is the first to demonstrate that Taf1 participates in B cell differentiation. However, this result emphasizes that further studies should be performed to assess the effect of Taf1 on B cell development and differentiation in animals and humans.

Our results indicated that candidate members of the gut microbiota, provided as feed additives, might microbiologically regulate B cell development in the bursa of birds. Future studies are needed to block the bursal duct ligation, which is the connection between the bursa and the gut, and consider the direct delivery of an appropriate microbiologic or antibiotic to birds, or injection into the bursal lumen immediately, in order to investigate the effects of microbiota stimulation on the intestines. Studies should also provide meaningful insights into B cell differentiation and correlations between B cells in the bursa and the gut microbiota.

**FIG 4** Legend (Continued)
are the predominant cell types in each cluster. (B and C) UMAP plot of the B cells identified by using *CHB6* in the bursa. UMAP plots of B cells (16 clusters) and non-B cells (6 clusters) from 21 organ tissues are shown. Each dot represents one cell, and each color-coded region represents one cell cluster. (D) Large and small B cell populations identified in panel B based on cell size. (E to G) Percentages of B, large B, and small B cell populations under antibiotic treatment. (H) Expression of differentiation-related genes between large and small B cells.

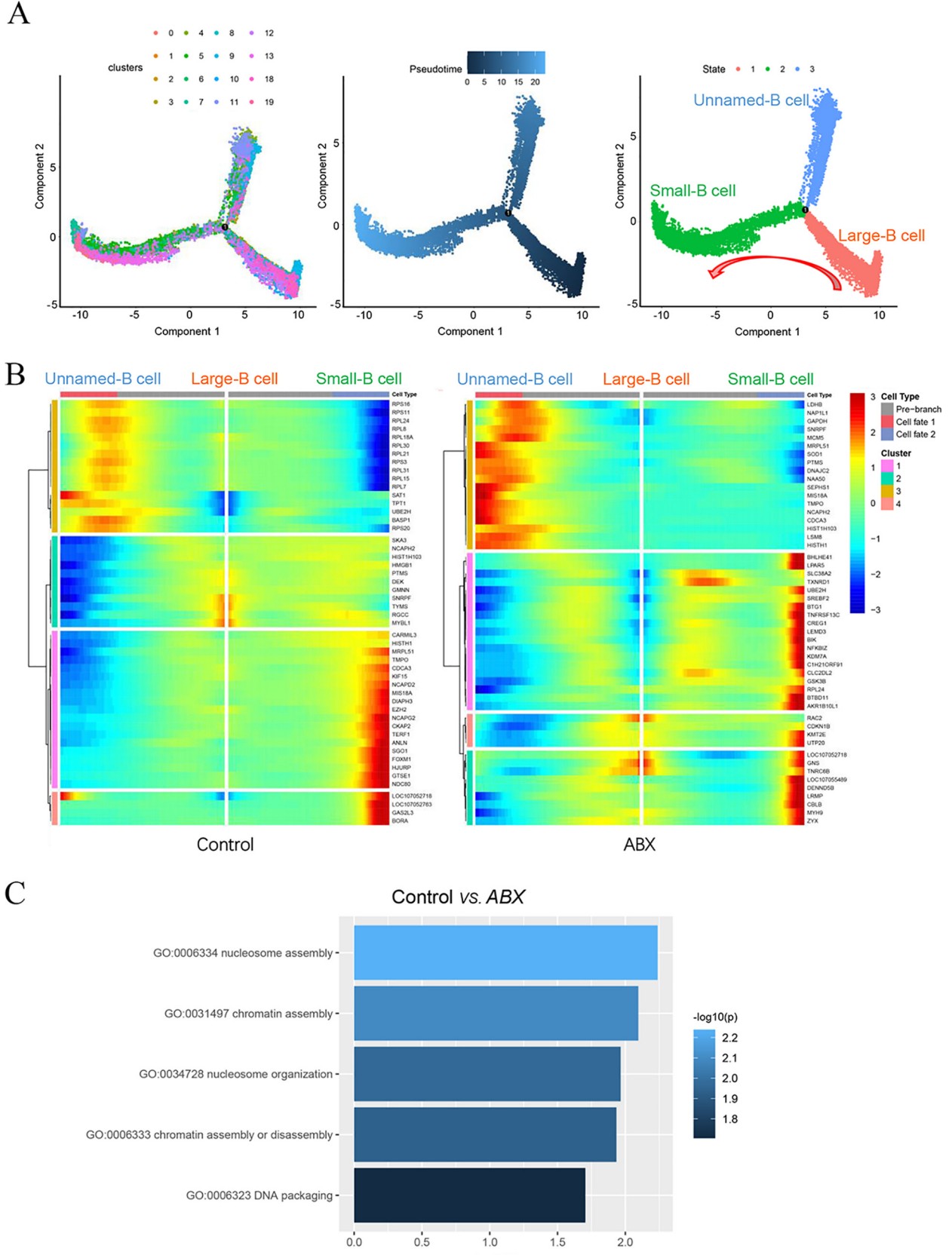

**FIG 5** Reconstruction of the developmental trajectory and pseudotime analysis of bursal B cells. (A) Uniform manifold approximation and projection (UMAP) plot of B cell clusters in tests defined by single-cell RNA sequencing (scRNA-seq) analysis. Pseudotime is colored in a

In summary, our study characterized the gut microbiota, B cells, and three immunoglobulins during the posthatch development of birds and identified several candidate taxa in the cecal contents that were most obviously associated with B cells. We provide a rich resource and cell-cell cross talk model supporting B cell differentiation from the bursa *in vitro* at single-cell resolution. We also identified Taf1 as a new pivotal regulator of B cell differentiation. These findings might enhance our understanding of the role of the gut microbiota in B cell differentiation and serve as a basis for developing novel strategies that can modulate B cell differentiation to prevent diseases.

## MATERIALS AND METHODS

**Animal study and experimental design.** Huiyang Bearded chickens (Chinese Yellow broiler breeders) used in the current study were cared for and used according to the guidelines of Guangdong Province on the review of welfare and ethics of laboratory animals, and the protocol was approved by the Guangdong Province Administration Office of Laboratory Animals. The birds were fed mostly with a corn and soybean meal-based diet formulated to meet the nutritional requirements. All chickens were allowed *ad libitum* feeding on a starter diet (200 g/kg of body weight crude protein (CP) and 2,900 kcal metabolic energy (ME)/kg) from hatching to 35 days of age, followed by feeding on a grower diet (180 g/kg CP and 2,950 kcal ME/kg). The first experiment was conducted on 60 1-day-old broilers for 154 days. Six birds were randomly selected for sample collection on days 1, 5, 9, 21, 35, 49, 70, 112, 140, and 154. For antibiotic treatment experiments, 80 1-day-old broilers were randomly divided into two equal groups. The birds were fed a standard diet. The broilers in the antibiotic treatment group were free to drink water supplemented with a mix of penicillin (200 mg/L), metronidazole (200 mg/L), and vancomycin (100 mg/L) through the whole experimental period. Samples were collected on days 1, 9, 49, and 140. The bursae of the chickens were dissected and weighed; the bursal index was calculated by dividing the organ weight by the chicken body weight. The cecum digesta were collected and stored at −80°C until further analysis; each BF was immediately rinsed in phosphate-buffered saline (PBS) and prepared for histological, immunohistochemical, and flow cytometric analyses.

**Histology and immunohistochemistry.** Portions of each bursa were collected and fixed in PBS containing 10% neutral buffered formalin. Paraffin-embedded sections (5 $\mu$m) were dewaxed, rehydrated, and stained with hematoxylin and eosin. Consecutive sections were also used for immunohistochemistry with mouse anti-chicken antibodies and detection reagents (Southern Biotech, Birmingham, AL). The antibodies used were against Bu-1$\alpha$ (catalog number 8395-02), chicken IgM (catalog number 8310-01), chicken IgA (catalog number 8330-01), and chicken IgY (catalog number 6100-01). The mean signal density was determined as the average integrated optical density/single lymphoid follicle by using Image-Pro Plus (version 6; Media Cybernetics, Rockville, MD).

**Flow cytometric analysis.** The BF was collected, minced, and filtered through a 70-$\mu$m nylon cell strainer (BD Falcon, San Jose, CA) to obtain a single-cell suspension stained with fluorescein isothiocyanate (FITC)-conjugated mouse anti-chicken Bu-1 antibody (catalog number 8395-31; Southern Biotechnology) in an ice bath for 30 min. After being stained, the cells were analyzed using a FACSCalibur flow cytometer (Becton, Dickinson, Palo Alto, CA) and Cell Quest software (Becton, Dickinson, Franklin Lakes, NJ).

**Microbial genomic DNA extraction and 16S rRNA gene sequencing.** The intestinal microbial composition of cecal samples from the birds was investigated. Total cecal microbial genomic DNA was obtained using our previously described method (35). The primers used for the amplification of the 16S rRNA gene V4 region were as follows: forward primer 5′-AYTGGGYDTAAAGNG-3′ and reverse primer 5′-TACNVGGGTATCTAATCC-3′. The melting temperature was 55°C, and 25 PCR cycles were performed. After being mixed and purified with a GeneJET gel extraction kit (Thermo Scientific), the PCR products were used to construct sequencing libraries according to the protocol provided by Illumina (San Diego, CA). The validated libraries were sequenced on an Illumina MiSeq platform, and 250-bp paired-end reads were generated (Personalbio Ltd., Shanghai, China).

**Microbiome sequencing data analysis.** Raw sequence reads were preprocessed to trim the adapter contamination reads, low-quality reads ($>$20% base quality [$>Q_{20}$]), and reads with any ambiguous base to obtain clean reads by using QIIME (v1.8.0). Clean paired-end reads with at least 10-bp overlaps were merged into tags via Connection Overlapped Pair-End (COPE) software (v1.2.1) (36). Bacterial tags were clustered into operational taxonomic units (OTUs) at 97% sequence identity by USEARCH (v10.0.240), and OTU taxonomic classification was conducted using Mothur based on the Ribosomal Database Project (RDP) database (37). Chao1, Simpson, and Shannon indices were calculated using Mothur (v1.31.2) (38), and the rarefaction curves were drawn using R software (v3.4.1). Beta diversity analysis based on the weighted UniFrac distance was conducted using QIIME software (v1.8.0). PCoA was performed using the vegan package, and the resulting distance matrices were visualized using R (v3.4.1). The correlations between microbes and immunohistochemistry factors were calculated using the Spearman algorithm; significant correlations ($P < 0.05$) were selected, and the heatmap was drawn using the pheatmap (v1.0.12) package in R (v3.4.1).

**FIG 5** Legend (Continued)
gradient from dark to light blue. The start and end of pseudotime are indicated by dark blue and light blue, respectively. (B) Heatmap representing significant differentially expressed genes (DEGs) between the control and antibiotic (ABX) treatment groups. The trajectory is colored by three cell states (large B cell, small B cell, and unnamed B cell). (C) Gene Ontology terms of differentially expressed genes in the B cell cluster from the control group versus the ABX group.

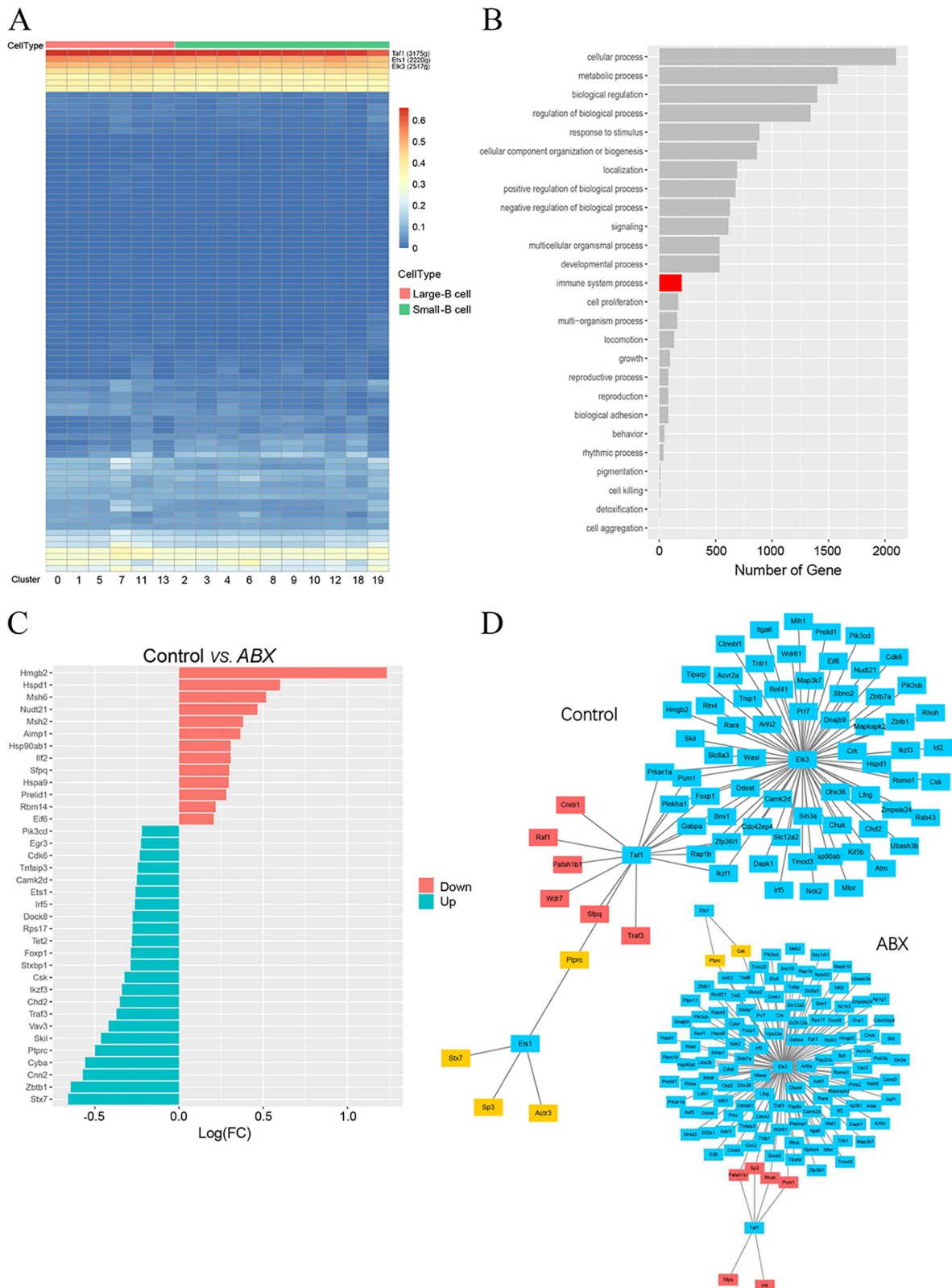

**FIG 6** Core regulons and regulatory differentially expressed genes (DEGs) in B cell differentiation. (A) SCENIC analysis predicts three transcription factors (TFs), namely, Taf1, Ets1, and Elk3, as central hubs governing the B cell state. TF regulon activities were quantified using AUCell. (B) Summary of KEGG pathway enrichment analysis of the biological process, cellular component, and molecular function categories. (C) Gene Ontology terms of DEGs related to the immune system from the control group versus the antibiotic (ABX) treatment group. Log(FC), log fold change. (D) A gene regulatory network (GRN) is a collection of regulatory interactions between three TFs and their target genes.

**Single-cell RNA sequencing.** Bursal tissue was collected from the birds in the control and antibiotic treatment groups on days 1, 9, 49, and 140. Single-cell suspensions were prepared from the isolated tissues. Debris-free suspensions with >80% viability were suitable for scRNA-seq. The suspensions were loaded onto a Chromium single-cell controller instrument (10× Genomics, Pleasanton, CA) to generate single-cell gel beads in emulsions (GEMs). scRNA-seq libraries were generated using Chromium single-cell 3 reagent version 3.1 kit (10× Genomics) according to the manufacturer's guidelines. Single-cell GEMs were then incubated with reverse transcription reagents, including several primers for cell and transcript barcoding for cDNA synthesis, followed by cDNA amplification and library construction. The libraries were sequenced on a DNBSEQ-T1 platform (BGI-Shenzhen, Shenzhen, China).

**Single-cell RNA-seq data preprocessing.** Sequencing data were processed using CellRanger (version 5.0.1; 10× Genomics, USA) with default settings. They were aligned to Genome Reference Consortium chicken build 6a (GRCg6a) by using STAR (39), and abnormal cells were uniformly screened in all data sets based on the CellRanger pipeline and quality control.

**scRNA-seq data analysis.** The output of CellRanger was imported into Seurat (v3.2.1) for downstream analysis (40). Cells with >200 genes detected of RNA or with >5% mitochondrial genes expressed were removed. Subsequently, the data were normalized, and 2,000 highly variable features (genes) were selected for linear dimensional reduction. Principal-component analysis was performed on a scaled gene sequence matrix, followed by uniform manifold approximation and projection (UMAP) for nonlinear dimensionality reduction. DEGs in clusters were identified using the following criteria: (i) only.pos = TRUE, (ii) min.pct = 0.25, and (iii) logfc.threshold = 0.25. Marker genes for the individual clusters were identified using the FindAllMarkers function in Seurat, and only significantly upregulated markers were selected. Single-cell trajectories were analyzed using the R package Monocle2 (v2.14.0) to reduce dimensionality and sort the cell matrix and gene expression (41).

**SCENIC analysis.** SCENIC is a computational method to infer gene regulatory networks (GRNs) from single-cell RNA-seq data (42). Seven representative samples were selected for SCENIC analysis. First, 20,000 cells were randomly selected from the seven samples to identify the sets of genes coexpressed with transcription factors by using GENIE3. Each coexpression module was analyzed using RcisTarget *cis*-regulatory motifs to identify putative direct binding targets. Modules with significant motif enrichment of the correct upstream regulator were retained. Next, the activity of each of the regulators in each cell in the seven samples was scored using AUCell. The resulting binary activity matrix was used for downstream analysis.

**Statistical analysis.** Statistical analyses were performed using Prism software (version 6.0c; GraphPad Software, La Jolla, CA), UniFrac, and Metastats. Comparisons between two groups were performed using Student's $t$ test. Pairwise community distances were determined using the weighted UniFrac algorithm. Taxon abundances at the phylum and genus levels were statistically compared between groups by using Metastats. Differences in UniFrac distances for pairwise comparisons among groups were determined using Student's $t$ test and the Monte Carlo permutation test with 1,000 permutations; they were also visualized using box-and-whisker plots. The significance of differences in the microbiota structures between groups was assessed by permutational multivariate analysis of variance (PERMANOVA) using the R package vegan. Spearman correlations between species abundances and immune factors were calculated using the R (v3.4.1) correlation test function and the estimated $P$ value. A $P$ value of <0.05 and a correlation of >0.6 were considered statistically significant.

**Data availability.** The sequencing data from this study have been deposited in the CNGB Sequence Archive (CNSA) (https://db.cngb.org/cnsa/) of the China National GenBank Database (CNGBdb) (https://db.cngb.org/) under accession numbers CNP0002855 and CNP0002822 (43, 44). Additional data from this study are contained in the article and its supplemental material.

## SUPPLEMENTAL MATERIAL

Supplemental material is available online only.
**SUPPLEMENTAL FILE 1**, PDF file, 1.9 MB.

## ACKNOWLEDGMENTS

This work was supported by the National Natural Science Foundation of China (grant number 32172687); the Science and Technology Program of Guangdong Province, People's Republic of China (grant number 2020B0202160009); the Special Fund for Scientific Innovation Strategy-Construction of High Level Academy of Agriculture Science (grant numbers R2020PY-JX006 and 202107TD); the Modern Agricultural Science and Technology Innovation Alliance of Guangdong (grant number 2019KJ106); and the CARS-Meat-Type Chicken (grant number CARS-41).

We declare that we have no conflict of interest.

Jian Ji and Hao Qu conceived and designed the experiments. Jian Ji, Jiaheng Cheng, Huangtao Lei, and Yushan Yuan performed the experiments. Jian Ji, Huangtao Lei, Jiaheng Cheng, Jianwei Chen, Xinyu Yi, Fang Zhao, Peng Chen, Jingyi Hi, Dingming Shu, and Chenglong Luo analyzed the data. Jian Ji wrote the paper. Chunlin Xie revised the manuscript. All authors discussed the results and approved the manuscript.

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
