## [Reviewer comments · Microbiology Spectrum]

Microbiology Spectrum

B lymphocyte development in the bursa of Fabricius of young broilers is influenced by the gut microbiota

Jianheng Cheng, Huangtao Lei, Jianwei Chen, Xinyu Yi, Fang ZHAO, Yushan Yuan, Peng Chen, Jingyi He, Chenglong Luo, Dingming Shu, Hao Qu, and Jian Ji

Corresponding Author(s): Jian Ji, State Key Laboratory of Livestock and Poultry Breeding, Guangdong Key Laboratory of Animal Breeding and Nutrition, Institute of Animal Science, Guangdong Academy of Agricultural Sciences

Review Timeline:

Submission Date:	November 22, 2022
Editorial Decision:	December 12, 2022
Revision Received:	January 5, 2023
Accepted:	January 12, 2023

Editor: Yunhe Fu

Reviewer(s): Disclosure of reviewer identity is with reference to reviewer comments included in decision letter(s). The following individuals involved in review of your submission have agreed to reveal their identity: Qianyun Xi (Reviewer #2)

Transaction Report:

DOI: <https://doi.org/10.1128/spectrum.04799-22>

December 12, 2022

Prof. Jian Ji

State Key Laboratory of Livestock and Poultry Breeding, Guangdong Key Laboratory of Animal Breeding and Nutrition, Institute of Animal Science, Guangdong Academy of Agricultural Sciences
No. 1, Dafeng one Street, Wushan Road, Institute of Animal Science
Guangzhou
China

Re: Spectrum04799-22 (B lymphocyte development in the bursa of Fabricius of young broilers is influenced by the gut microbiota)

Dear Prof. Jian Ji:

Link Not Available

Sincerely,

Yunhe Fu

Journals Department
Reviewer comments:

Reviewer #1 (Comments for the Author):

The manuscript by Cheng et al. showed that the relationship between B cells in the bursa of Fabricius and gut microbiota in young chicks, which is interesting. However, the paper could be made better, but this will require substantial revisions. Major comments:

1. In abstract:

(1) Change "the effect of the microbiota on mediating B cell development throughout life and ..." to "the effect of the microbiota on mediating B cell development", "throughout life" can be deleted.

(2) Change "... Bu-1 α (a marker for B cells) ..." to "... B cell marker Bu-1 α ..."

2. In Materials and methods:

(1) What is kind of the broiler chicken should be mentioned.

(2) How long for the antibiotic treatment?

(3) Check 2.5 Microbiome sequencing data analysis, "The correlations between microbes and immunohistochemistry factors were calculated using the "Spearman" algorithm; significant correlations ($P < 0.05$) were selected, and the heatmap was drawn using the pheatmap (v1.0.12) package by R (v3.4.1)", is the pheatmap the right one?

3. In 3.2, "The pattern of changes in the gut microbiota and B cells in chickens of all ages was similar on different days", here the diversity (or other index) of gut microbiota and the development of B cells should be clear.

4. Much more relative elements should be cited in Discussion

Reviewer #2 (Comments for the Author):

Comments to the Author

The manuscript describes the relationship between gut microbiota and bursal B cell development. The topic will contribute to the field of poultry microbiology, and will be of interest in a broader research domain. The experimental and methodological of the study is rigorous, and the data is available. However, minor issues require attention before a recommendation can be made.

Introduction:

Line 54-56 The author want to discuss the relationship between the gut microbiota and bursal B cells in birds. However, the reference cited by author is redundancy. The author should rephrase or delete the sentence.

Line 57-58 In this section the author emphasized the importance of gut microbiota in immune system. However, the literature has focused on human and mice, the author should provide more evidence that gut microbiota plays a critical role in poultry immune system regulation. (Clavijo 2018. Poult Sci. DOI: 10.3382/ps/pex359)

Materials and methods:

Line 79-80 Composition of the basal experimental diet for of animals should be provided.

Results:

Please check throughout the manuscript whether the quotation is correct which is a serious problem. Line 262 One picture appears twice for different figures.

Line 269 Please check out the Figure.

Line 277-279 The results didn't show that gut microbiota mediated B cells development in the bursa at earlier stages. The author should explain.

Discussion and conclusion

Line 368 Delete the superscript.

Line 405-406 The manuscript didn't focus on the influence of PAMP on B cell differentiation. The author should explain why they draw this conclusion.

Staff Comments:

Preparing Revision Guidelines

Please return the manuscript within 60 days; if you cannot complete the modification within this time period, please contact me. If you do not wish to modify the manuscript and prefer to submit it to another journal, please notify me of your decision immediately so that the manuscript may be formally withdrawn from consideration by Microbiology Spectrum.

Rebuttal letter

RE: Spectrum04799-22

Dear Dr. Fu,

Thank you very much for your positive handling of our manuscript “B lymphocyte development in the bursa of Fabricius of young broilers is influenced by the gut microbiota”. We greatly appreciate the helpful comments from the reviewers. We have now performed additional data analysis according to the reviewers’ suggestions and integrated the corresponding changes (in blue) into our revised manuscript. Our revisions and responses to the reviewers’ comments are detailed in appended Point-by-point responses to the reviewers. If any questions about our revised manuscript, please don’t hesitate to contact me.

Best Wishes

Yours sincerely,

Jian Ji, PhD

Professor, Institute of Animal Science

Guangdong Academy of Agricultural Sciences

Guangzhou 510640

CHINA

Reviewer comments:

Reviewer #1 (Comments for the Author):

The manuscript by Cheng et al. showed that the relationship between B cells in the bursa of Fabricius and gut microbiota in young chicks, which is interesting. However, the paper could be made better, but this will require substantial revisions. Major comments:

1. In abstract:

(1) Change "the effect of the microbiota on mediating B cell development throughout life and ..." to "the effect of the microbiota on mediating B cell development", "throughout life" can be deleted.

Response: We thank the reviewer for raising this point. We have deleted the "throughout life".

(2) Change "... Bu-1 α (a marker for B cells) ..." to "... B cell marker Bu-1 α ..."

Response: We have changed "Bu-1 α (a marker for B cells)" to "B cell marker Bu-1 α ".

2. In Materials and methods:

(1) What is kind of the broiler chicken should be mentioned.

Response: Thanks for the question. We have provided the information about the kind of broiler chicken in revised manuscript in line 77.

(2) How long for the antibiotic treatment?

Response: Sorry for missing the detail. We have added the time of antibiotic treatment in lines 88-90.

(3) Check 2.5 Microbiome sequencing data analysis, "The correlations between microbes and immunohistochemistry factors were calculated using the "Spearman" algorithm; significant correlations ($P < 0.05$) were selected, and the heatmap was drawn using the pheatmap (v1.0.12) package by R (v3.4.1)", is the pheatmap the right one?

Response: After checking the information, pheatmap (v1.0.12) is software for drawing heatmap.

3. In 3.2, "The pattern of changes in the gut microbiota and B cells in chickens of all ages was similar on different days", here the diversity (or other index) of gut microbiota and the development of B cells should be clear.

Response: As the reviewer's said, the pattern of changes in alpha diversity of gut microbiota and B cell marker Bu-1 α expression. We have added the relative information in 3.2.

4. Much more relative elements should be cited in Discussion.

Response: We have cited some relative elements including the literature in revised version.

Reviewer #2 (Comments for the Author):

Comments to the Author

The manuscript describes the relationship between gut microbiota and bursal B cell development. The topic will contribute to the field of poultry microbiology, and will be of interest in a broader research domain. The experimental and methodological of the study is rigorous, and the data is available. However, minor issues require attention before a recommendation can be made.

Introduction:

Line 54-56 The author want to discuss the relationship between the gut microbiota and bursal B cells in birds. However, the reference cited by author is redundancy. The author should rephrase or delete the sentence.

Response: We thank the reviewer for raising the issue. We have deleted the sentence.

Line 57-58 In this section the author emphasized the importance of gut microbiota in immune system. However, the literature has focused on human and mice, the author should provide more evidence that gut microbiota plays a critical role in poultry immune system regulation. (Clavijo 2018. Poult Sci. DOI: 10.3382/ps/pex359)

Response: Thanks for this advice. We have cited the relative reference about influence of the microbiota on poultry immunity in current lines 66-69.

Literatures:

1. Dirkjan Schokker, Alfons J M Jansman, Gosse Veninga, Naomi de Bruin, Stephanie A Vastenhouw, Freddy M de Bree, Alex Bossers, Johanna M J Rebel, Mari A Smits. Perturbation of microbiota in one-day old broiler chickens with antibiotic for 24 hours negatively affects intestinal immune development. BMC Genomics. 2017 Mar 20;18(1):241. doi: 10.1186/s12864-017-3625-6.
2. Viviana Clavijo, Martha Josefina Vives Flórez. The gastrointestinal microbiome and its association with the control of pathogens in broiler chicken production: A review. Poult Sci. 2018 Mar 1;97(3):1006-1021. doi: 10.3382/ps/pex359.

Materials and methods:

Line 79-80 Composition of the basal experimental diet for of animals should be provided.

Response: Thanks for the suggestion. We have provided composition of the basal experimental diet for of animals in current lines 82-84.

Results:

Please check throughout the manuscript whether the quotation is correct which is a serious problem. Line 262 One picture appears twice for different figures.

Response: Sorry for the mistake. After checking, we found that two pictures (FigS3 and Fig S8) appear twice for different figures. We have uploaded the right figure.

Line 269 Please check out the Figure.

Response: Thanks for the reminder. After double check the Figure, we have revised the Fig 4 to the Fig 3 in current manuscript.

Line 277-279 The results didn't show that gut microbiota mediated B cells development in the bursa at earlier stages. The author should explain.

Response: We thank the reviewer for raising the issue. We agree with the reviewer's thoughtful comments. We have revised the sentence and move to the in current lines 297-301.

Discussion and conclusion

Line 368 Delete the superscript.

Response: We have deleted the superscript.

Line 405-406 The manuscript didn't focus on the influence of PAMP on B cell differentiation. The author should explain why they draw this conclusion.

Response: Sorry for the inaccurate expression. We have deleted the sentence and revised some sentences.

January 12, 2023

Prof. Jian Ji

State Key Laboratory of Livestock and Poultry Breeding, Guangdong Key Laboratory of Animal Breeding and Nutrition, Institute of Animal Science, Guangdong Academy of Agricultural Sciences
No. 1, Dafeng one Street, Wushan Road, Institute of Animal Science
Guangzhou
China

Re: Spectrum04799-22R1 (B lymphocyte development in the bursa of Fabricius of young broilers is influenced by the gut microbiota)

Dear Prof. Jian Ji:

Your manuscript has been accepted, and I am forwarding it to the ASM Journals Department for publication. You will be notified when your proofs are ready to be viewed.

Sincerely,

Yunhe Fu
Editor, Microbiology Spectrum
